# Mitochondrial ATP transporter depletion protects mice against liver steatosis and insulin resistance

Joonseok Cho[1], Yujian Zhang[2], Shi-Young Park[3], Anna-Maria Joseph[4], Chul Han[4], Hyo-Jin Park[4], Srilaxmi Kalavalapalli[5], Sung-Kook Chun[6], Drake Morgan[7], Jae-Sung Kim[6], Shinichi Someya[4], Clayton E. Mathews[1], Young Jae Lee[3], Stephanie E. Wohlgemuth[8], Nishanth E. Sunny[5], Hui-Young Lee[3], Cheol Soo Choi[3,9], Takayuki Shiratsuchi[2], S. Paul Oh[10] & Naohiro Terada[1]

Non-alcoholic fatty liver disease (NAFLD) is a common metabolic disorder in obese individuals. Adenine nucleotide translocase (ANT) exchanges ADP/ATP through the mitochondrial inner membrane, and *Ant2* is the predominant isoform expressed in the liver. Here we demonstrate that targeted disruption of *Ant2* in mouse liver enhances uncoupled respiration without damaging mitochondrial integrity and liver functions. Interestingly, liver specific Ant2 knockout mice are leaner and resistant to hepatic steatosis, obesity and insulin resistance under a lipogenic diet. Protection against fatty liver is partially recapitulated by the systemic administration of low-dose carboxyatractyloside, a specific inhibitor of ANT. Targeted manipulation of hepatic mitochondrial metabolism, particularly through inhibition of ANT, may represent an alternative approach in NAFLD and obesity treatment.

[1] Department of Pathology, University of Florida College of Medicine, Gainesville, Florida 32610, USA. [2] Otsuka Maryland Medicinal Laboratories, Rockville, Maryland 20850, USA. [3] Korea Mouse Metabolic Phenotyping Center, Lee Gil Ya Cancer and Diabetes Institute, Gachon University School of Medicine, Incheon 406-840, Korea. [4] Department of Aging, University of Florida College of Medicine, Gainesville, Florida 32610, USA. [5] Department of Medicine, University of Florida College of Medicine, Gainesville, Florida 32610, USA. [6] Department of Surgery, University of Florida College of Medicine, Gainesville, Florida 32610, USA. [7] Department of Psychiatry, University of Florida College of Medicine, Gainesville, Florida 32610, USA. [8] Department of Animal Sciences, University of Florida College of Medicine, Gainesville, Florida 32611, USA. [9] Endocrinology, Internal Medicine, Gachon University Gil Medical Center, Incheon 405-760, Korea. [10] Department of Physiology, University of Florida College of Medicine, Gainesville, Florida 32610, USA. Correspondence and requests for materials should be addressed to N.T. (email: terada@pathology.ufl.edu).

Adenine nucleotide translocase (ANT), also called ADP/ATP carrier (AAC), is a solute carrier that specifically exchanges ADP and ATP through the mitochondrial inner membrane by an antiport mechanism[1–4]. Thus, the molecule is essential for coupled respiration leading to ATP production in mitochondria. Mice have three paralogous ANT genes, Ant1, Ant2 and Ant4 (ref. 5). Ant1 is highly expressed in skeletal and cardiac muscle, and its disruption leads to exercise intolerance and cardiomyopathy[6,7]. Ant4 is exclusively expressed during germ cell meiosis and is essential for male meiosis and fertility[8–10]. Ant2, in contrast, is ubiquitously expressed in all somatic tissues, thus it is thought to play a critical role in supplying energy in the whole body[11]. When we recently characterized systemic Ant2-hypomorphic mice, contrary to our prediction, only limited organs and cell types were affected by Ant2 depletion[12]. Most strikingly, liver development was not overtly affected by Ant2 depletion despite the fact that the liver expresses Ant2 almost exclusively among all available Ant paralogs. Since the Ant2 hypomorphic mice died within 3–4 weeks after birth likely due to a gastric paresis complication[12], we were not able to assess the effect of Ant2 deficiency in the liver over a long period of time. Kokoszka et al. previously described that the mice having Ant2 conditional knockout (cKO) in the liver along with systemic Ant1 KO background survived for over 12–15 months; however, the authors did not describe the detailed liver phenotype or effects on metabolism of the mice[13]. To investigate the long-term influence of Ant2 depletion in the liver, we recently generated our own liver-specific Ant2 cKO mice. Here we show that Ant2-depleted liver is indeed intact over 10 months with increases in mitochondrial mass and uncoupled respiration. Interestingly, liver specific Ant2 cKO mice are leaner and resistant to hepatic steatosis, obesity and insulin resistance under a high fat diet. Targeted manipulation of hepatic mitochondrial metabolism, particularly through inhibition of ANT, may become an alternative approach in non-alcoholic fatty liver disease (NAFLD) and obesity treatment.

## Results

**Liver specific Ant2 cKO mice are viable but leaner.** When the Ant2 gene is disrupted in hepatocytes using Albumin/Cre (Alb/Cre) mice, the Ant2 protein became undetectable in the liver (but unaffected in other tissues) (Fig. 1a,b). A compensational increase in Ant1 protein expression was not observed in the cKO liver (Fig. 1b). The Alb/Cre + ;Ant2$^{fl/Y}$ (cKO) mice were overtly healthy and normal over 10 months, and showed no differences from control mice (Alb/Cre − ;Ant2$^{fl/Y}$) except a noticeably leaner phenotype (Fig. 1c). Indeed, the body weight was significantly lower than controls from 10 weeks of age (Fig. 1d), and fat mass was decreased approximately two thirds when body composition was measured by Time-Domain Nuclear Magnetic Resonance (TD-NMR) at 16 weeks (Fig. 1e). Appetite assessed by food intake, and physical activity assessed by grip strength, locomotion and wheel running activity, were not statistically different between control and cKO mice (16–18 weeks old, Supplementary Fig. 1).

**Ant2 is dispensable for liver homeostasis.** The control and cKO livers were similar in size, but the cKO liver was noticeably more red possibly implying a reduced fat content in the organ (see triglycerides data below) (Fig. 2a). On histological analysis, the cytoplasm of cKO hepatocytes was uniformly eosinophilic by H&E staining compared with control hepatocytes (Fig. 2b, upper panel). No increase in cell death or inflammatory changes was detected. Transmission electron microscopy showed a clear

increase in mitochondria in the cKO hepatocytes (Fig. 2b, middle panel), which was confirmed by multiphoton microscopy analysis after staining mitochondria with Rhodamine-123 (Fig. 2b, lower panel). Indeed, mitochondrial contents per gram of liver were significantly increased over threefold in the cKO liver (Fig. 2c). PGC1-α expression was also increased in the cKO liver indicating enhanced mitochondrial biogenesis (Fig. 2d). In blood chemistry, normal levels of ALT, AST, total serum protein, albumin and bilirubin were observed (Supplementary Table 1) indicating no sign on liver inflammation or dysfunction. These data indicate that Ant2 is dispensable for liver development and homeostasis. It should be noted here that AMPK phosphorylation status in the liver was not significantly altered by Ant2 depletion (Supplementary Fig. 2a), indicating either that residual ATP transport capacity in mitochondria plus the substantial increase in mitochondrial numbers may supply sufficient ATP to hepatocytes, or alternative pathways such as glycolysis provide the bulk of ATP required for liver homeostasis. In addition, there were no changes in expression of catalase and SOD2 in the Ant2 cKO liver indicating no increases in reactive oxygen species in the organ (Supplementary Fig. 2b).

**Uncoupled respiration in Ant2-depleted liver mitochondria.** Since Ant1 is expressed at a low level and Ant4 is not expressed at all in the liver[12], deletion of Ant2 should deprive the cells of nearly all ADP/ATP exchange activity. Indeed, when we examined ADP/ATP exchange activity in isolated liver mitochondria, the exchange activity was decreased over 96% in Ant2-KO liver mitochondria (Fig. 3a). In contrast, transport of other solutes such as pyruvate was not changed in the cKO mitochondria (Supplementary Fig. 3a), indicating mitochondrial inner membrane integrity was preserved. Similarly, the outer membrane integrity was not altered in the cKO mitochondria, when assessed by oxidation of reduced cytochrome c (Supplementary Fig. 3b). Of interest, despite the profound decrease in ADP/ATP exchange capacity, the Ant2 cKO liver mitochondria sustained similar membrane potential and basal respiration rate (State 2 in the presence of glutamate; Fig. 3b,c). Addition of ADP to isolated mitochondria is known to maximize coupled respiration (State 3), and indeed it increased the respiration rate over threefold in control mitochondria (Fig. 3c, right panel). However, with significant reduction in the ADP/ATP exchange capacity (Fig. 3a), Ant2-depleted mitochondria were not responsive to ADP addition at all. ADP addition even decreased the respiration rate as indicated by the State3/State2 ratio being lower than a value of 1 (Fig. 3c). To confirm this finding, we also examined oxygen consumption of liver mitochondria with a Seahorse extracellular flux analyzer (Fig. 3d). Similarly, mitochondria isolated from Ant2-depleted liver failed to respond by consuming oxygen after addition of ADP. These data indicate that respiration in Ant2-deleted liver mitochondria is not coupled with ADP/ATP exchange. Taken together, Ant2-deleted liver mitochondria likely sustain respiration and membrane potential putatively with an increase in uncoupled respiration. Indeed, when compared with wild-type mitochondria, Ant2-depleted mitochondria showed a higher phosphorylation-independent proton leak in the presence of succinate, rotenone, oligomycin and various doses of malonate, a competitive inhibitor of succinate dehydrogenase (complex II of the electron transport system) (Fig. 3e, Supplementary Fig. 3c). A reduced sensitivity to malonate in Ant2-deleted mitochondria may be due to increases in respiratory complex proteins (Supplementary Fig. 3d).

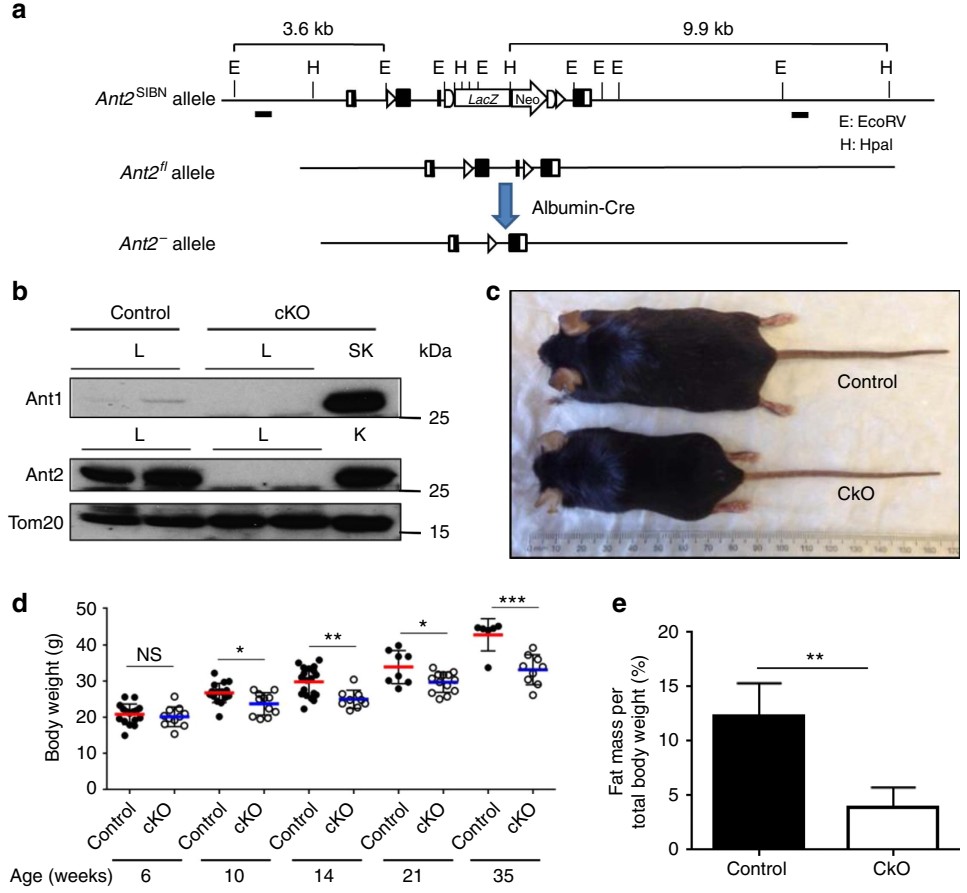

**Figure 1 | Conditional deletion of the *Ant2* gene in the murine liver.** (**a**) Schematic diagram of liver-specific *Ant2* gene disruption strategy. (**b**) Immunoblot analysis for Ant1, Ant2 and TOM20 expression in mitochondrial lysates isolated from liver (L), skeletal muscle (SK) and kidney (K) of control (*Alb/Cre − ;Ant2^{fl/Y}*) and cKO (*Alb/Cre + ;Ant2^{fl/Y}*) mice. (**c**) Representative phenotypes of control and cKO mice (20 weeks old). (**d**) Body weight of 6- to 35-week-old mice (*$P < 0.05$, **$P < 0.01$, ***$P < 0.001$). (**e**) TD-NMR analysis of fat mass in 16 weeks old mice ($n = 12$; *$P < 0.05$ by *t*-test, error bars = s.d.).

**Liver specific Ant2 cKO mice are resistant to steatosis**. The increase in mitochondrial mass with predominantly uncoupled respiration will induce the liver to consume nutrients such as glucose and fatty acid with a greater speed but without producing energy. Indeed, serum glucose, cholesterol and insulin in *Ant2* cKO liver were decreased by ∼30%, 37% and 56%, respectively, when compared with the control (Supplementary Table 1). In addition, ketone bodies were increased in the serum (Supplementary Table 1). This led us to the hypothesis that liver-specific *Ant2*-deficient mice will be tolerant to overnutrition. To test the hypothesis, we fed the mice a high-fat (40%)/high-fructose (20%) diet for 8 weeks, which is known to induce fatty liver in mice. As shown in Fig. 4a–c, the control mice developed severe fatty liver on gross examination, histology and liver triglyceride levels. In contrast, mice with *Ant2* cKO livers were almost completely resistant to developing fatty liver changes under this regimen.

**Liver specific Ant2 cKO mice are resistant to obesity**. We also used a high-fat (60%) diet for 4 weeks, which is known to induce obesity and insulin resistance in the mice. As shown in Fig. 5a, increased body weight was mitigated in the cKO mice. In addition, *Ant2* cKO mice showed a lower fat mass and reduced white adipose tissue weight at the experimental endpoint (4 weeks; Fig. 5b,c). To investigate the cause of reduced body

weight, we measured whole-body energy expenditure using an animal metabolic monitoring system. Although not significant, the energy expenditure of *Ant2* cKO mice tended to be higher when compared with that of control mice (Fig. 5d). In addition, fasting plasma glucose and HbA1c concentrations were significantly reduced in cKO mice after 4 weeks of high-fat diet treatment[7]. Since high-fat diet is also known to induce insulin resistance in mice, we performed hyperinsulinemic-euglycemic clamp analysis after 4 weeks of the high-fat diet regimen. The cKO mice showed significantly higher insulin sensitivity, represented by a high glucose infusion rate to maintain the euglycemia under insulin infusion, indicating that *Ant2* liver cKO is protective against changes that are common in pre-diabetes (Fig. 5f). These data indicate that a liver-specific deletion of Ant2 is particularly effective to prevent fatty liver and also able to moderate development of systemic obesity and its subsequent complications.

**Enhanced uncoupled respiration is independent of Ucp2**. The concept of using mitochondrial uncouplers to treat NAFLD has been recently demonstrated in the literatures[14,15]; however, here we show, for the first time, that depletion of an ATP transporter exerts a similar and powerful effect. This was unexpected since the ANT molecule itself has been implicated, in previous reports, to have uncoupling capacities[16–19]. To date,

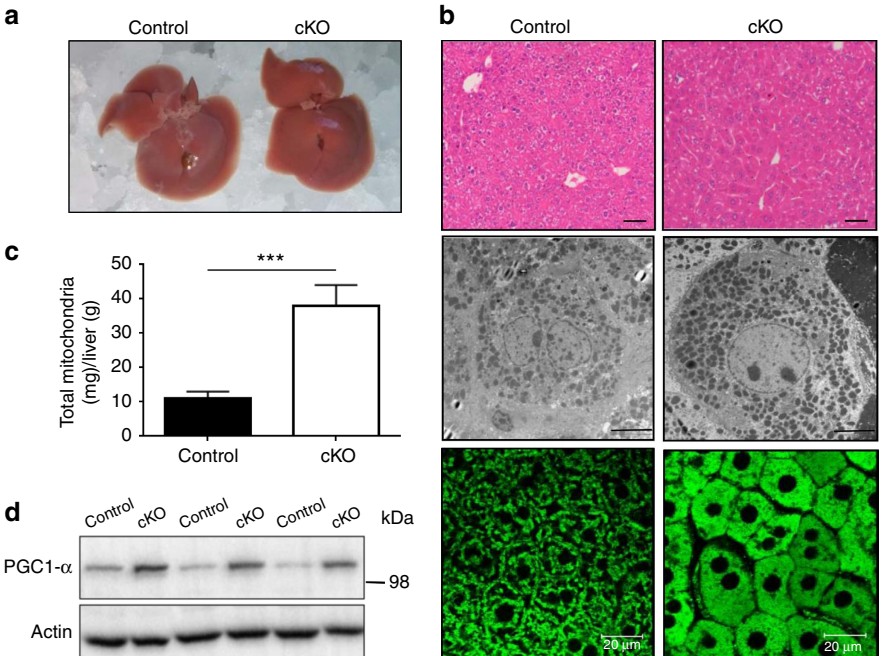

**Figure 2 | Increases in mitochondrial numbers in the *Ant2* cKO liver.** (**a**) Representative images of liver gross appearance (*n* = 10), (**b**) Representative images of H&E staining, Scale bar; 40 μm (upper panel), transmission electron microscopy analysis Scale bar; 5 μm (middle panel) and multiphoton microscopy analysis after the liver was perfused with Rhodamine-123 Scale bar; 20 μm (lower panel; *n* = 5), (**c**) mitochondrial mass per gram of liver (*n* = 5; ***P < 0.001 by *t*-test, error bars = s.d.), and (**d**) immunoblot analysis of the liver.

we do not know what molecule mediates the uncoupled proton dissipation in Ant2-depleted liver mitochondria. The uncoupling protein 1 (Ucp1) and Ucp3 are not expressed in the liver[20]. Ucp2 is expressed in the liver and was increased approximately threefold by *Ant2* disruption (Supplementary Fig. 4a). However, when we generated liver-specific *Ant2*/*Ucp2*-double KO mice the mitochondrial phenotypes remained unchanged as *Ant2*-single KO mice (Supplementary Fig. 4b). Furthermore, *Ucp2* conditional KO did not reverse the fatty liver prevention effect by Ant2 depletion (Supplementary Fig. 4c). These data indicate that Ucp2 does not likely play a role in the increased uncoupled respiration in Ant2-deleted liver mitochondria. The adenine nucleotide translocator and the mitochondrial F$_1$Fo ATP synthase are thought to be in close contact as part of a supermolecular complex[21]. It is conceivable that in cells lacking ANT and in the absence of ADP in the mitochondrial matrix, that ATP synthase may become leaky or favour ATP hydrolysis rather than synthesis, and thereby transport protons out of the mitochondrial matrix, leading to uncoupling. However, addition of oligomycin (Fig. 3d) or cyclosporine A (Supplementary Fig. 3e) did not change OCR, implying no involvement of ATP synthase or cyclophilin-mediated permeability transition in the uncoupled respiration here.

**Chemical inhibition of ANT mitigates fatty liver changes.** Finally, to test whether chemical inhibition of ANT can mimic a liver protective effect of *Ant2* deletion, we administered a low dose of carboxyatractyloside (CATR) intraperitoneally during the last 2 weeks of the 8 weeks-high fat/high fructose regimen. As shown in Supplementary Fig. 5, systemic CATR administration partially improved fatty liver changes in gross examination, histologically and at the level of liver triglyceride accumulation. These data imply that chemical ANT inhibition may have a potential for NAFLD treatment particularly in the development

of compounds that become active only within hepatocytes and/or a system to deliver the agents specifically to the liver. Alternatively, RNA interference-mediated ANT reduction in the liver may be a practical option since there is considerable clinical and pre-clinical experience in the use of viral vectors for liver directed gene therapy[22–24]. Since humans have an additional ANT paralog, ANT3, that is expressed ubiquitously as ANT2 (ref. 25), simultaneous knockdown of ANT2 and ANT3 may be necessary in this case although their overlapping function is still controversial[26].

NAFLD affects up to 30% of people in the United States and is one of the most common metabolic disorders closely associated with obese and overweight individuals[27]. NAFLD may progress to steatohepatitis and over time to irreversible cirrhosis. Moreover, it has been recently shown that NAFLD is an independent risk factor for cardiovascular complications including coronary artery disease and stroke[27,28]. Altering diet is the most straightforward method to prevent NAFLD and obesity, but success rates of dietary programs are unfortunately low. Furthermore, some types of obesity cannot be prevented by limiting food intake alone due to pathologically altered lipid metabolism. The present study demonstrates that targeted manipulation of hepatic mitochondrial metabolism, particularly through inhibition of ANT, could become an alternative solution to treat NAFLD and obesity, leading health concerns in the modern world.

## Methods
**Mice.** Mice bearing *Ant2* conditional KO allele (*Ant2$^{fl}$*) were generated as we described previously[12]. Liver-specific conditional *Ant2* KO mice were developed by mating a heterozygous female mouse bearing *Ant2$^{fl}$* allele with a male mouse containing a transgene of Cre recombinase controlled by an albumin promotor (Alb-Cre). Selected male mice with positive for the *Ant2$^{fl}$* allele and Alb-Cre (*Alb/Cre +;Ant2$^{fl/Y}$*) were used at 14–20 weeks of age in all experiments. The littermate *Alb/Cre −;Ant2$^{fl/Y}$* mice were used as controls. *Ucp2$^{fl/fl}$* male mice were purchased from The Jackson Laboratory (022394) to generate *Ant2/Ucp2* double conditional KO mice. Genotyping of the mice was performed by polymerase chain

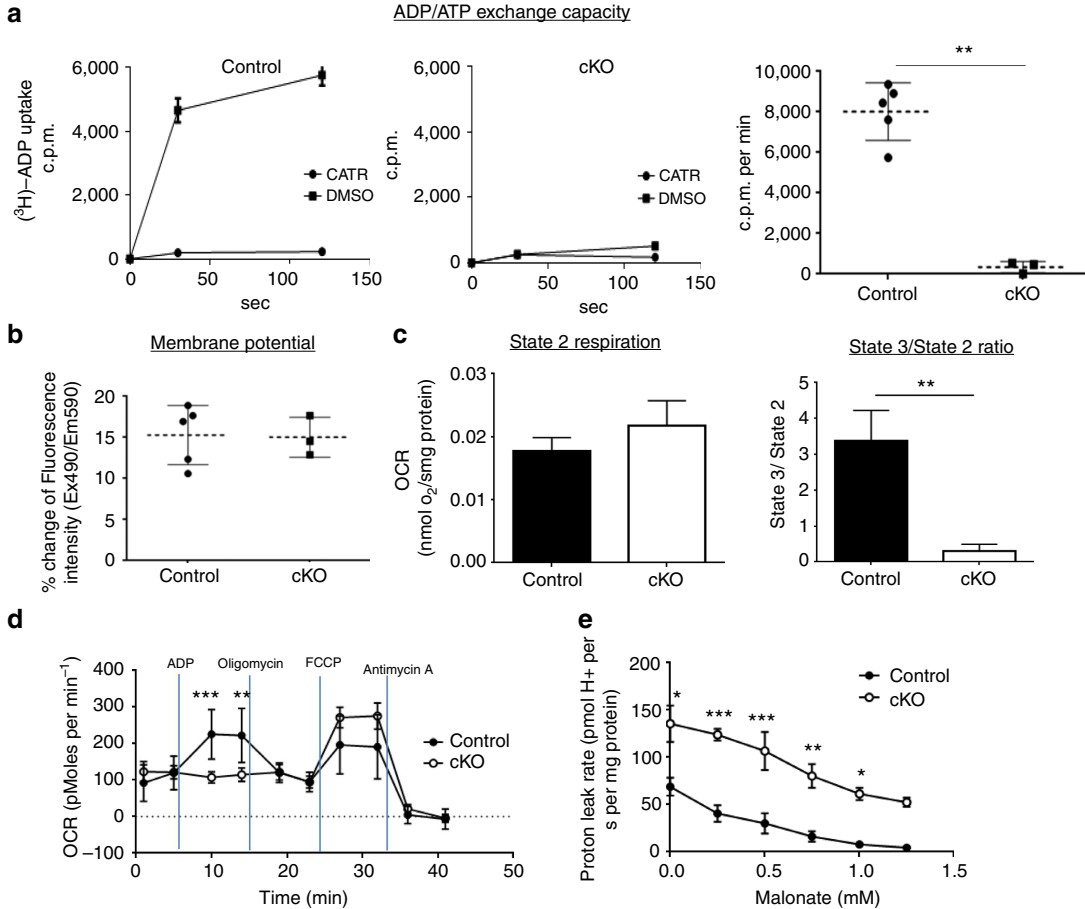

**Figure 3 | Characterization of *Ant2*-deleted liver mitochondria. (a)** ADP/ATP exchange capacity measured by [3H]ADP uptake in the presence or absence of the ANT specific inhibitor, carboxyatractyloside (CATR). Left two panels show representative data from control and *Ant2* cKO liver mitochondria, while the far right graph shows the average of multiple experiments ($n = 5$; **$P < 0.01$ by $t$-test, error bars = s.d.). **(b)** Membrane potential assessed by JC1 dye. **(c)** Mitochondrial oxygen consumption rate (OCR) measured using a Hansatech Oxygraph unit. The state 2 respiration represents the basal respiration rate under glutamate while the state 3 respiration rate was measured after the addition of ADP ($n = 5$; **$P < 0.01$ by $t$-test, error bars = s.d.). **(d)** Mitochondrial OCR assessed with XF24 extracellular flux analyzer (Seahorse) under sequential treatment of ADP, oligomycin, FCCP, and Antimycin A ($n = 5$; **$P < 0.01$, ***$P < 0.001$ by $t$-test, error bars = s.d.). **(e)** Proton leak rate of isolated liver mitochondria was assessed with Oroboros O2k respirometer in the presence of succinate, rotenone, oligomycin and increasing concentrations of malonate ($n = 5$, *$P < 0.05$, **$P < 0.01$, ***$P < 0.001$ by $t$-test, error bars = s.d.).

reaction (PCR) of genomic DNA. Primer sequences used were 5′-ATACAGCT CGGTGGTAGAGCATTA-3′ (F8) and 5′-AGCACAGGCATTGACTGGAGA ACA-3′ (R8) for control and *Ant2^fl* alleles yielding 193 and 289 bp bands, respectively. Cre recombinase gene was identified with 5′-GAAGCAGAA GCTTAGGAAGATGG-3′ (Forward) and 5′-TTGGCCCCTTACCATAACG-3′ (Reverse) primers. *Ucp2* was assessed by 5′-ACCAGGGCTGTCTCCAAGCA GG-3′ (Forward), 5′-AGAGCTGTTCGAACACCAGGCCA-3′ (Reverse1), and 5′-TAGAGGAGGGTGGTGTTCCAGCTC-3′ (Reverse2) primers yielding a 270 bp band for WT, a 380 bp band for *Ucp2^fl* and a 516 bp band for *Ucp^−*. Control and *Ant2* cKO littermates were fed regular chow (Harlan, 7912), high fat/high fructose diet (Research Diets, Inc., D09100301), or high-fat diet (Research Diets Inc., D12492). All animal experiments were conducted after approval by the Institutional Animal Care and Use Committee of the University of Florida and Center of Animal Care and Use (CACU) at Lee Gil Ya Cancer and Diabetes Institute, Gachon University. *Ant2^fl* mouse line will be available from The Jackson Laboratory as JAX#029482.

**Grip strength.** Forelimb grip strength was determined using an automated grip strength meter (Columbus Instruments, Columbus, OH). The mouse was grasped by the base of the tail and suspended above a grip ring. After approximately 3 s, the mouse was gently lowered towards the grip ring and allowed to grasp the ring with its forepaws. The remainder of the mouse's body was quickly lowered to a horizontal position, and the animal's tail was pulled until grasp of the ring was broken. The mean force in grams was determined with a computerized electronic

pull strain gauge that was fitted directly to the grasping ring. Maximal force obtained from three trials was used as the dependent measure.

**Locomotor activity.** Animals were placed in activity chambers (43 × 43 cm, Med Associates, Inc.) for 3-min sessions. Total distance traveled (centimeter) and spatial location (time in 'center' or 'margin') were tracked with an overhead camera and Ethovision XT 7.0 software (Noldus Information Technology Inc., Wageningen, The Netherlands).

**Time-domain nuclear magnetic resonance.** Body composition was assessed using TD-NMR in restrained but fully conscious mice (TD-NMR Minispec, Bruker Optics, The Woodlands, TX). The MiniSpec identifies three components of body composition (fat, free body fluid, and lean tissue in grams) by acquiring and analysing TD-NMR signals from all protons in the sample area. Scans were acquired by placing mice into a cylindrical restrainer (90-mm diameter and ∼250-mm length), fitted with a screw top that tightens to the length of the mice. The restrainer was then inserted into the analyser. The total scan time was ∼1 min, and the average of two scans was used as the final measurement.

**Quantitative RT-PCR.** Total RNA was isolated from cells by RNA queousTM kit (Ambion, Austin, TX, USA) and cDNA was synthesized by high capacity cDNA reverse transcription kit (Applied Biosystems). For *Ant1* and *Ant2*, quantitative RT-PCR was performed as we described previously[29]. For *UCP1*, *UCP2* and *UCP3*, SYBR Green (Applied Biosystems) was used according to the manufacturer's

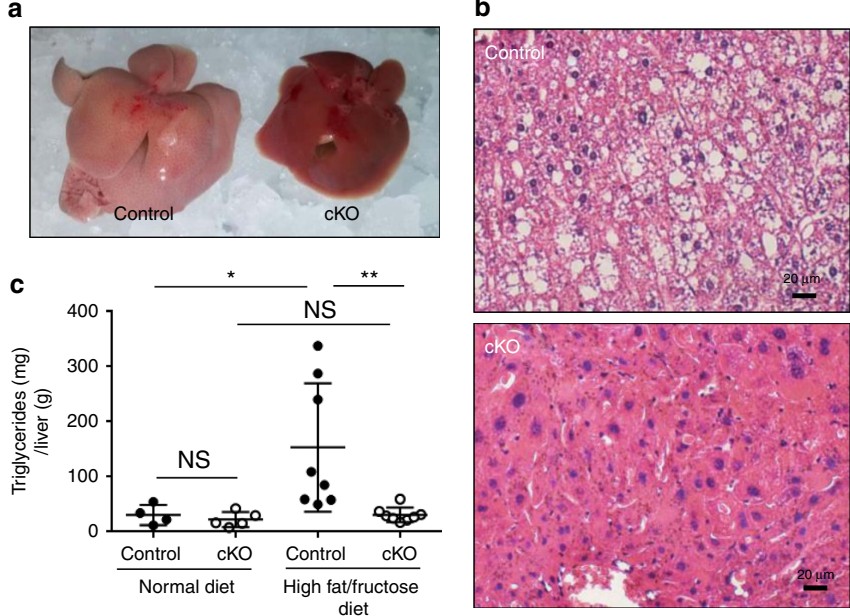

**Figure 4 | *Ant2*-deletion in the liver prevents fatty liver development under a high fat/fructose diet.** Control and *Ant2* cKO mice at the age of 6 weeks were subjected to a high fat (40%)/high fructose (20%) diet for 8 weeks. (**a**) Representative images of liver gross appearance (*n* = 8). (**b**) Representative images of liver histology analysis with H&E staining (*n* = 10). Scale bar; 20 μm. (**c**) Total triglyceride levels in the liver under a normal or high fat/fructose diet (*n* > 5, *P < 0.05, **P < 0.01 by *t*-test, error bars = s.d.).

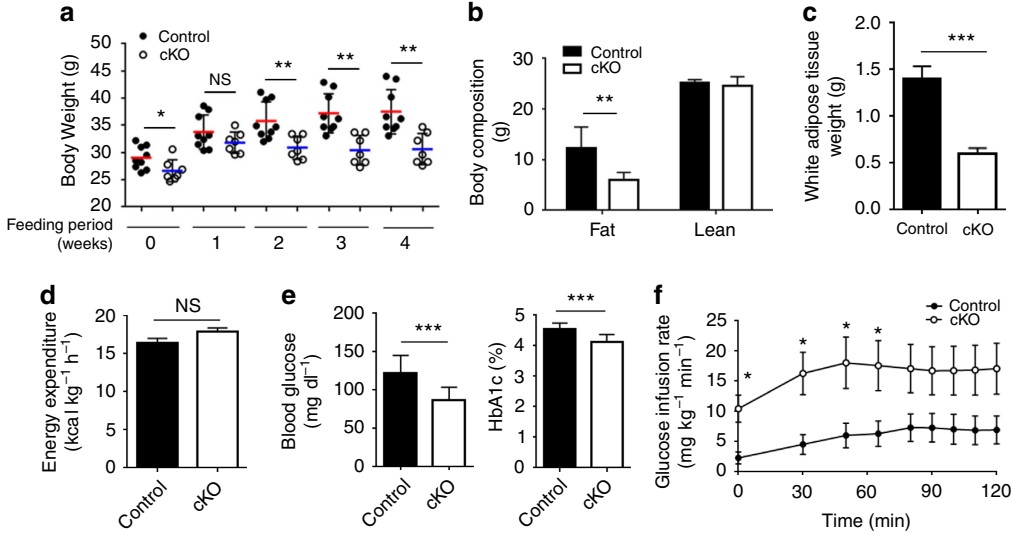

**Figure 5 | *Ant2* deletion in the liver mitigates obesity under a high fat diet.** Control and *Ant2* cKO mice at the age of 28 weeks were subjected to a high fat diet (60%) for 4 weeks. (**a**) Body weight changes under the high fat diet regimen (*n* > 9, *P < 0.05, **P < 0.01 by *t*-test, error bars = s.d.). (**b**) Fat and lean mass assessed by TD-NMR analysis (*n* = 10, **P < 0.01 by *t*-test, error bars = s.d.). (**c**) Weight of white adipose tissue (*n* = 10, ***P < 0.001 by *t*-test, error bars = s.d.). (**d**) Energy expenditure assessed by a comprehensive laboratory animal monitoring system (CLAMS). (**e**) Plasma levels of fasting glucose and HbA1c (*n* = 10, ***P < 0.001 by *t*-test, error bars = s.d.). (**f**) Glucose infusion rate during hyperinsulinemic-euglycemic clamp (*n* = 5, *P < 0.05 by *t*-test, error bars = s.d.).

instructions with primers: UCP1 5′-TATCATCACCTTCCCGCTG-3′ (Forward) and 5′-GTCATATGTTACCAGCTCTG-3′ (Reverse), UCP2 5′-CTGGCAGGTAG CACCACAG-3′ (Forward) and 5′-AAAGGTGCCTCCCGAGATT-3′ (Reverse), UCP3 5′-CCTCTACGACTCTGTCAAGC-3′ (Forward) and 5′-GACAGGGG AAGTTGTCAGTA-3′ (Reverse).

**Immunoblot analysis.** Immunoblot analysis was performed as we described previously[30]. Rabbit polyclonal anti-ANT-1 antibody and rabbit monoclonal anti-β-actin antibody were from Sigma-Aldrich (St Louis, MO, USA, SAB2105530, 1:1,000) and Cell Signaling (Danvers, MA, USA, 4970, 1:5,000), respectively.

Rabbit polyclonal anti-ANT2 antibody was generated by immunization with KLH-conjugated N-terminus ANT2 peptide (NH-TDAAVSFAKDFLAG-COOH) at Lampire Biological Laboratories (Ottsville, PA, USA) and purified with protein G (GE Healthcare, Buckinghamshire, UK) and antigen-conjugated columns. The ANT2 antibody was diluted to 1:200 as being used. COX IV (4850, 1:1,000) and VDAC (4866, 1:1,000) antibodies were from Cell Signaling (Danvers, MA, USA). Tom20 antibody was from Santa Cruz Biotechnology (sc-11415, 1:500), Catalase and SOD2 antibodies were from Sigma (St. Louis, MO, USA, C0979, 1:1,000) and Abcam (Cambridge, UK, ab16956, 1:1,000), respectively. Uncropped scans of the critical immunoblots are shown in Supplementary Fig. 6.

**Histology.** Littermate mice were killed at the ages of 10, 15 and 20 weeks. Freshly collected livers were fixed in 4% paraformaldehyde overnight. Sections of the block embedded in paraffin were stained with hematoxylin and eosin.

**Mitochondria preparation.** Mitochondria were isolated by differential centrifugation method. Briefly, after mice were killed, liver was surgically removed and weighed. The liver was homogenized by Dounce homogenizer in IBc buffer (10 mM Tris-MOPS, 1 mM EGTA, 0.2 M sucrose, pH 7.4). The homogenate was centrifuged at 600 g for 10 min in a refrigerated centrifuge, and the supernatant was again centrifuged at 7,000 g for 10 min. The precipitated pellets were washed once with the same buffer and re-suspended in respiration buffer (120 mM KCl, 5 mM KH$_2$PO$_4$, 1 mM EGTA, 3 mM HEPES, pH 7.4). Protein concentration was determined by Coomassie Plus Protein Assay (Pierce).

**ADP/ATP exchange assay.** The ADP/ATP exchange assay was performed on a filter-plate (glassfiber FB, Millipore). Briefly, mitochondria (50 μg) in 25 μl of respiration buffer was pre-incubated with 50 μl of carboxyatractyloside (CATR, 1 μM as final concentration) or DMSO as vehicle control for 10 min on ice. The ADP/ATP exchange was initiated by adding 25 μl of $^3$H-ADP (5 μM and 2 μCi ml$^{-1}$). The reaction was stopped by adding 50 μl of 2 μM CATR after 0.5 and 2 min. Filter-plate was vacuumed and washed twice with IBc buffer on a manifold (Millipore). Supermix (PerkinElmer) of 100 μl was added into each well. The radioactivity in each well was measured by MicroBeta after overnight incubation.

**Mitochondria membrane potential.** The membrane potential of isolated mitochondria was measured with JC-1 dye (Mitochondria Staining Kit, Sigma). Mitochondria (40 μg) in 80 μl of 1 × JC1 assay buffer were incubated with 20 μl of 0.5 μg ml$^{-1}$ of JC-1 in the same buffer at room temperature. The change of the fluorescence (Ex490/Em590) was recorded during 10 min period. FCCP at 10 μM was used as a positive control.

**Pyruvate uptake assay.** Mitochondria were suspended at 4 mg ml$^{-1}$ in respiration buffer (120 mM KCl, 5 mM KH$_2$PO$_4$, 1 mM EGTA, 3 mM HEPES, pH 7.4). The suspension of 250 μl was transferred to 750 μl of the same buffer (pH 6.8) containing [2-$^{14}$C]pyruvate (200 μM and 0.75 μCi ml$^{-1}$) and 25 μM malate as the final concentration. After 1.5 or 3 min, 200 μl was taken and loaded onto 0.5 ml of oil layer made with AR20 and AP100 (4:1) and spun down immediately (1 min at 12,000 g). After removing the oil layer, the tip of the tube was clipped with a cutter. The pellet was dissolved with 100 μl of SOLVABLE (Perkin Elmer) and transferred to 3 ml of UltraGold scintillation solution for counting. The radioactivity was calculated by subtracting the background level acquired in the presence of pyruvate carrier-specific inhibitor-UK5099 (100 μM).

**Mitochondrial outer membrane integrity test.** The integrity of mitochondrial outer membrane was measured by Cytochrome c Oxidase Assay Kit (Sigma) according to manufacturer's instruction. Briefly, two parallel samples of the mitochondrial suspension were diluted to 0.1 mg ml$^{-1}$ with either 1 × Enzyme Dilution Buffer or the Enzyme Dilution Buffer containing 1 mM n-dodecyl β-D-maltoside (total cytochrome c oxidase activity). The samples were incubated on ice for 15 min. Mix 5 μl of mitochondria solution with 15 μl of Ferrocytochrome c Substrate Solution in 180 μl of 1 × Assay Buffer and immediately start to record the kinetic change of OD$_{550}$. The membrane integrity of mitochondria was calculated based on its activity as a percentage of total cytochrome c oxidase activity.

**Blood chemistry.** Peripheral blood was collected via heart puncture and submitted to the core of Animal Care Service at the University of Florida for blood chemistry by Hemavet HV950FS (Drew Scientific, Dallas, TX, USA).

**Triglyceride levels.** Triglyceride concentrations were determined using an analytical kit from Sigma (St.Louis, MO).

**Oxygen consumption rate.** Mitochondrial O$_2$ consumption was measured using a Clark-type electrode coupled to an Oxygraph unit (Hansatech), an Oxygraph-2k respirometer (O2k, Oroboros, Innsbruck, Austria), and an XF24 extracellular flux analyzer (Seahorse Bioscience, North Billerica, MA, USA). For O2k mesurements, respiration media (MiR05) containing 0.5 mM EGTA, 3 mM MgCl$_2$*6H$_2$O, 60 mM potassium lactobionate, 20 mM taurine, 10 mM KH$_2$PO$_4$, 20 mM HEPES, 110 mM sucrose, and 1 g l$^{-1}$ fatty acid free BSA was used. Mitochondrial proton leak rate was assessed by incremental addition of malonate from 0 to 1,250 μM in the presence of 2 μg ml$^{-1}$ oligomycin, 0.5 μM rotenone, and 10 mM succinate.

Effect of ADP and oligomycin on mitochondrial respiration was assessed using a Seahorse XF24 respirometer. Mitochondria (10 μg per well in triplicates) were placed in basal respiration buffer (10 mM sodium pyruvate, 2 mM malate, 2 mM glutamate, and 5 mM succinate) and sequentially treated with 1 mM ADP,

1 μg ml$^{-1}$ oligomycin, 1 μM FCCP, and 2 μM antimycin A in final concentrations. In some experiments, 1 μM cyclosporine A was used. OCR was measured two times with 2 min interval after treating with the chemicals.

**Intravital multiphoton microscopy.** To visualize polarized mitochondria, livers were labelled with Rhodamine 123 (Rd-123), a mitochondrial membrane potential (ΔΨm)-sensitive fluorophore. A 24-gauge catheter was inserted into the portal vein and Rd-123 in Krebs–Ringer–HEPES (KRH) buffer (15 mM NaCl, 5 mM KCl, 2 mM CaCl$_2$, 1.2 mM MgSO$_4$, 1 mM KH$_2$PO$_4$, and 25 mM HEPES, at pH 7.4) was infused for 10 min (50 ml of 10 μmol l$^{-1}$ per animal). The liver was gently withdrawn from the abdominal cavity and placed over a glass coverslip on the stage of a Zeiss LSM510 equipped with a multi-photon microscope. Images of green fluorescing rhodamine 123 were collected with a 40 × water-immersion objective lens. Rhodamine 123 was excited with 780 nm from a Chameleon Ultra Ti-Sapphire pulsed laser (Coherent, Inc., Santa Clara, CA), and images were collected through a 500- to 550-nm band pass filter.

**Mouse metabolic studies.** To measure whole-body energy metabolism in high fat diet-fed mice, a comprehensive laboratory animal monitoring system (CLAMS; Columbus Instruments) was used for 4 days (2 days of acclimation followed by 2 days of measurement).

**Hyperinsulinemic–euglycemic clamp.** After an overnight fasting, [3-$^3$H]-glucose (HPLC purified; Perkinelmer, USA) was infused at a rate of 0.05 μCi min$^{-1}$ for basal 2 h to assess the basal glucose turnover. Following the basal period, hyperinsulinemic-euglycemic clamp was conducted for 150 min with a primed/continuous infusion of human insulin (21.4 mU per kg during priming and 3 mU per kg per min during infusion, Eli Lilly), while plasma glucose was maintained at basal concentrations as previously described[31] with slight modification. To estimate insulin-stimulated whole body glucose fluxes, [3-$^3$H]-glucose was infused at a rate of 0.1 μCi min$^{-1}$ throughout the clamps.

**Statistical analysis.** Statistical significance between groups was determined using either Student's t-tests or analysis of variance in Prism GraphPad Software. (*$P < 0.05$, **$P < 0.01$ and ***$P < 0.001$)

**Data availability.** The data sets generated during and/or analysed during the current study are available from the corresponding author on reasonable request.

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

## Acknowledgements

We thank Drs Ken Cusi, Mike Kilberg, Chelsey Simmons, Kyle Rowe, Seh-Hoon Oh, Bryon Petersen, Matthew Merritt, Yasemin Sakarya and Philip Scarpe for helpful discussion and technical advice, and Dr Katherine Santostefano for critical reading of the manuscript. This research was supported in part by National Institutes of Health Grants, HD060474 to N.T., DK079879 and DK090115 to J.-S.K., DK074656 to C.E.M., a research grant from Otsuka Maryland Medicinal Laboratories to N.T., and Korea Healthcare Technology R&D Project (HI14C1135) and National Research Foundation grant (NRF-2014R1A1A1002009, 2014M3A9D5A01073886) to H.-Y.L. and C.S.C.

## Author contributions

J.C. performed research, analysed data, designed research and wrote the paper; Y.Z., S.-Y.P., A.-M.J., C.H. H.-J.P., S.K., S.-K.C. and D.M. performed research; Y.J.L., S.S., C.E.M. and J.-S.K. performed research and analysed data; T.S., S.E.W., N.E.S., H.-Y.L., C.S.C. and S.P.O. designed research and analysed data; and N.T. designed research, analysed data and wrote the paper

## Additional information

**Competing financial interests:** The authors declare no competing financial interests.

DOI: 10.1038/ncomms16143    OPEN

# Corrigendum: Mitochondrial ATP transporter depletion protects mice against liver steatosis and insulin resistance

Joonseok Cho, Yujian Zhang, Shi-Young Park, Anna-Maria Joseph, Chul Han, Hyo-Jin Park, Srilaxmi Kalavalapalli, Sung-Kook Chun, Drake Morgan, Jae-Sung Kim, Shinichi Someya, Clayton E. Mathews, Young Jae Lee, Stephanie E. Wohlgemuth, Nishanth E. Sunny, Hui-Young Lee, Cheol Soo Choi, Takayuki Shiratsuchi, S. Paul Oh & Naohiro Terada

*Nature Communications* 8:14477 doi: 10.1038/ncomms14477 (2017); Published 16 Feb 2017; Updated 18 Aug 2017

In the References section of this Article, incorrect papers are cited for references 4, 5, 7, 9, 10, 13–20 and 29–31. In addition, in the Methods section, reference 12 should be cited instead of reference 30 following the statement 'Immunoblot analysis was performed as we described previously', and reference 30 should be cited instead of reference 31 following the statement 'while plasma glucose was maintained at basal concentrations as previously described'. The corrected references are listed below.

The list of corrections is as it follows.

● Ref #4 [Valianpour, F. *et al.* Monolysocardiolipins accumulate in Barth syndrome but do not lead to enhanced apoptosis. *J. Lipid Res.* **46**, 1182–1195 (2005)] is now [Monné, M. & Palmieri, F. Antiporters of the mitochondrial carrier family. *Curr. Top. Membr* **73**, 289–320 (2014)].

● Ref #5 [Gonzalez, I. L. Barth syndrome: TAZ gene mutations, mRNAs, and evolution. *Am. J. Med. Genet. A* **134**, 409–414 (2005)] is now [Rodić, N. *et al.* DNA methylation is required for silencing of ant4, an adenine nucleotide translocase selectively expressed in mouse embryonic stem cells and germ cells. *Stem Cells* **23**, 1314–1323 (2005)].

● Ref #7 [Winker, R. *et al.* Functional adrenergic receptor polymorphisms and idiopathic orthostatic intolerance. *Int. Arch. Occup. Environ. Health* **78**, 171–177 (2005)] is now [Esposito, L. A., Melov, S., Panov, A., Cottrell, B. A. & Wallace, D. C. Mitochondrial disease in mouse results in increased oxidative stress. *Proc. Natl Acad. Sci. USA* **96**, 4820–4825 (1999)].

● Ref #9 [Winker, R. *et al.* Endurance exercise training in orthostatic intolerance: a randomized, controlled trial. *Hypertension* **45**, 391–398 (2005)] is now [Brower, J. V., Lim, C. H., Jorgensen, M., Oh, S. P. & Terada, N. Adenine nucleotide translocase 4 deficiency leads to early meiotic arrest of murine male germ cells. *Reproduction* **138**, 463–470 (2009)].

● Ref #10 [Brinkman, J., de Nef, J. J., Barth, P. G. & Verschuur, A. C. Burkitt lymphoma in a child with Joubert syndrome. *Pediatr. Blood Cancer* **44**, 397–399 (2005)] is now [Lim, C. H., Brower, J. V., Resnick, J. L., Oh, S. P. & Terada, N. Adenine nucleotide translocase 4 is expressed within embryonic ovaries and dispensable during oogenesis. *Reprod. Sci.* **22**, 250–257 (2015)].

● Ref #13 [Polo, J. M. *et al.* Cell type of origin influences the molecular and functional properties of mouse induced pluripotent stem cells. in *Nat. Biotechnol.* 28, 848–855 (2010)] is now [Kokoszka, J. E. *et al.* The ADP/ATP translocator is not essential for the mitochondrial permeability transition pore. *Nature* **427**, 461–465 (2004)].

● Ref #14 [Mangat, J., Lunnon-Wood, T., Rees, P., Elliott, M. & Burch, M. Successful cardiac transplantation in Barth syndrome--single-centre experience of four patients. *Pediatr. Transplant.* **11**, 327–331 (2007)] is now [Perry, R. J. *et al.* Reversal of hypertriglyceridemia, fatty liver disease, and insulin resistance by a liver-targeted mitochondrial uncoupler. *Cell Metab.* **18**, 740–748 (2013)].

● Ref #15 [Huth, S., Jäger, D. & Barth, J. A young fireman candidate with an abnormal chest x-ray]. *Internist (Berl)* **48**, 532–534, 536 (2007)] is now [Tao, H., Zhang, Y., Zeng, X., Shulman, G. I. & Jin, S. Niclosamide ethanolamine-induced mild mitochondrial uncoupling improves diabetic symptoms in mice. *Nat. Med.* **20**, 1263–1269 (2014)].

● Ref #16 [Spencer, C. T. *et al.* Ventricular arrhythmia in the X-linked cardiomyopathy Barth syndrome. *Pediatr. Cardiol.* **26**, 632–637 (2005)] is now [Shabalina, I. G., Kramarova, T. V., Nedergaard, J. & Cannon, B. Carboxyatractyloside effects on brown-fat mitochondria imply that the adenine nucleotide translocator isoforms ANT1 and ANT2 may be responsible for basal and fatty-acid-induced uncoupling respectively. *Biochem. J.* **399**, 405–414 (2006)].

● Ref #17 [Tang, T. *et al.* Combined lifestyle modification and metformin in obese patients with polycystic ovary syndrome. A randomized, placebo-controlled, double-blind multicentre study. *Hum. Reprod.* **21**, 80–89 (2006)] is now [Andreyev A. Yu. *et al.* The ATP/ADP-antiporter is involved in the uncoupling effect of fatty acids on mitochondria. *Eur. J. Biochem.* **182**, 585–592 (1989)].

● Ref #18 [Schlame, M., Ren, M., Xu, Y., Greenberg, M. L. & Haller, I. Molecular symmetry in mitochondrial cardiolipins. *Chem. Phys. Lipids* **138**, 38–49 (2005)] is now [Andreyev A. Yu. *et al.* Carboxyatractylate inhibits the uncoupling effect of free fatty acids. *FEBS Lett.* **226**, 265–269 (1988)].

● Ref #19 [Soyka, M. *et al.* Treatment of alcohol withdrawal syndrome with a combination of tiapride/carbamazepine: results of a pooled analysis in 540 patients. *Eur. Arch. Psychiatry Clin. Neurosci.* **256**, 395–401 (2006)] is now [Lee, Y. S. *et al.* Increased adipocyte O2 consumption triggers HIF–1α, causing inflammation and insulin resistance in obesity. *Cell* **157**, 1339–1352 (2014)].

● Ref #20 [Finsterer, J., Stöllberger, C., Gaismayer, K. & Janssen, B. Acquired noncompaction in Duchenne muscular dystrophy. *Int. J. Cardiol.* **106**, 420–421 (2006)] is now [Chavin, K. D. *et al.* Obesity induces expression of uncoupling protein–2 in hepatocytes and promotes liver ATP depletion. *J. Biol. Chem.* **274**, 5692–5700 (1999)].

● Ref #29 [Huang, S. C. *et al.* Mitral annuloplasty in an infant with Barth syndrome and severe mitral insufficiency: first case report and determination of annular diameter. *J. Thorac. Cardiovasc. Surg.* **136**, 1095–1097 (2008)] is now [Brower, J. V. *et al.* Differential CpG island methylation of murine adenine nucleotide translocase genes. *Biochim. Biophys. Acta* **1789**, 198–203 (2009)].

● Ref #30 [Xu, Y., Sutachan, J. J., Plesken, H., Kelley, R. I. & Schlame, M. Characterization of lymphoblast mitochondria from patients with Barth syndrome. *Lab. Invest.* **85**, 823–830 (2005)] is now [Choi, C. S. *et al.* Continuous fat oxidation in acetyl-CoA carboxylase 2 knockout mice increases total energy expenditure, reduces fat mass, and improves insulin sensitivity. *Proc. Natl Acad. Sci. USA* **104**, 16480–16485 (2007)].

● Ref #31 [Das, R. *et al.* The British Cardiac Society Working Group definition of myocardial infarction: implications for practice. *Heart* **92**, 21–26 (2006)] should be removed.

The correct list of References is as follows:

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
