## [Peer review file · Nature Communications]

Reviewers' comments:

Reviewer #1 (Remarks to the Author):

This manuscript reports the effects of liver knockout of ANT2 on fatty liver phenotypes in mice. The results are pretty clear that loss of ANT2 leads to an overall lean phenotype in mice likely due to increased hepatic uncoupled respiration and likely better fat burning in the liver. This seems to be novel and is not what I expected. There is good therapeutic potential. The data are clearly presented and concisely summarized. I think that the conclusions are adequately treated as experimental. I'm satisfied that the conclusions are justified. The paper is well written. I have a few minor concerns.

The lean phenotype is very clear and there is a suggestion that the liver uncoupled respiration is increased. However, I don't think that it's well supported that this is due to overall increases in whole-body energy expenditure and this should be tested. This is the biggest question in my mind.

I have a minor question regarding the ANT1 blot. It almost appears that this protein is also diminished.

Reviewer #2 (Remarks to the Author):

A The question raised here is to explain the findings that the ANT deletion in liver has not the expected life threatening effects given the central role of the ANT in ATP supply. Mitochondria remain intact h on deletion ANT. Interestingly, the synthesis of mitochondria is even increased . Since respiration with substrates such as pyruvate is normal, an uncoupling has to be invoked. Since the membrane potential is not decreased the uncoupling has to be subtle. Uncoupling proteins are not involved. The nature of the uncoupling remains unknown .

A major disease related aspect of this paper are the fat metabolism changes associated with the ANT levels. Fatty liver disease on over-nutrition is suppressed by ANT deletion. A similar prevention of the fatty liver or even of general obesity has the injection of the inhibitor CAT in normal mice.

B. This is an interesting paper where paradoxical phenomena challenge conventional wisdom.

C ok

D ok

E The conclusions should be modified according to forthcoming results (see F)

F inhibitor CAT in normal mice.

The following amendments are suggested. In order to understand the status of the ANT deleted mitochondria the content of ATP synthase (AS) is needed. AS activity should be decrease in ANT depleted mitochondria. On the other hand AS might be responsible for the uncoupling. For example if ANT forms super-complexes with AS . The lack of ANT might destabilize the AS dimers which are known to form pores in the "permeability transition". Thus AS might play a key role for understanding the ANT deficient mitochondrial

metabolism

G. ok

H. ok

Reviewer #3 (Remarks to the Author):

Cho et al analyze hepatocyte-specific ANT2 null mice, which exhibit low body weight, protection from HFD induced adiposity, and protection from fatty liver. These phenotypes are associated with hepatocyte mitochondrial proliferation, abrogated ADP-stimulated respiration, and indirect evidence of uncoupled respiration that does not require UCP2. The authors suggest ANT2 may be a target for NAFLD. The studies are interesting, but would improve from modest experimental revision. More confirmation of phosphorylation-independent proton leak, driving increased substrate oxidation, is needed.

1. Uncoupled respiration has been supported, but not demonstrated, because mitochondrial number is increased, and state 2 respiration is normalized to mitochondrial mass. A respiration experiment in liver tissue or isolated hepatocytes should be performed to demonstrate increased state 2 respiration.
2. Formal measurements of proton leak should be made, e.g. as performed in Cell Metab. 2006;3:417-27. Membrane potential appears to be normal in ANT2KO. To confirm proclivity toward uncoupling, rates of membrane potential-dependent proton leak should be measured using escalating malonate concentrations (which collapse membrane potential)
3. The hypothesis that F1F0 ATPase may favor proton leak and ATP hydrolysis is attractive, and easy to assess using oligomycin. If it is not the F1F0 ATPase, then there are other potential candidates.
4. Liver is typically 20% of resting energy expenditure. It will almost certainly be much higher in these animals. At the very least, whole animal VO₂/energy expenditure must be measured.
5. The source of the Ant2fl allele is not clear. Is this new or the same as that in Nature 427, 461-465 (2004)? If it's the same allele, then Fig. 1a should move to supplemental. If it's a new construction, then more details should be given in the methods.
6. Give body weights on HFD.
7. The term pseudo-fasting should be eliminated. This is not at all the fasting state, even with weight loss and mild ketosis.

RESPONSE TO REVIEWERS

We do appreciate a thorough and positive review of the manuscript and very helpful suggestions provided by the reviewers. We have carefully considered each of the points raised by the reviewers, documented in the point-by-point responses to their concerns. We have addressed all the important criticisms and also provided the new data. Thanks to all the reviewers' helpful suggestions, we believe that the revised version of this manuscript has been significantly improved.

All changes in the text are highlighted in **yellow**. **Figures 3d, 3e and 5, Extended Data Figures 3c and 3d** contain newly added content.

Please find point-by-point responses to the reviewers' comments below.

Reviewer #1,

This manuscript reports the effects of liver knockout of ANT2 on fatty liver phenotypes in mice. The results are pretty clear that loss of ANT2 leads to an overall lean phenotype in mice likely due to increased hepatic uncoupled respiration and likely better fat burning in the liver. This seems to be novel and is not what I expected. There is good therapeutic potential. The data are clearly presented and concisely summarized. I think that the conclusions are adequately treated as experimental. I'm satisfied that the conclusions are justified. The paper is well written. I have a few minor concerns.

The lean phenotype is very clear and there is a suggestion that the liver uncoupled respiration is increased. However, I don't think that it's well supported that this is due to overall increases in whole-body energy expenditure and this should be tested. This is the biggest question in my mind.

We tested whole-body energy expenditure of the mice under a high fat diet using a CLAMS. Although it tended to be slightly higher in the *Ant2* cKO mice, there was no statistical difference. These new data are now included in the manuscript as **Fig. 5d**, and discussed in the text.

I have a minor question regarding the ANT1 blot. It almost appears that this protein is also diminished.

Yes, we have repeatedly seen the tendency that expression of Ant1 protein but not mRNA was decreased in the *Ant2*-knockout liver. We have not seen a similar phenomenon in the liver of the systemic *Ant2* hypomorphic mice (*Cell Death Differ* **22**, 1437, 2015) or the heart of *Ant2^{fl/y}-MHC6Cre* mice (not published). We initially thought that this could be due to MIM impairment potentially caused by the complete loss of Ant2 protein in the liver mitochondria. However, as shown in the manuscript, MIM integrity has been well preserved. At this moment, we do not know the reason. It should be noted, however, that there is a residual (<4%) activity of ADP/ATP exchange in the *Ant2* cKO liver mitochondria (Fig. 3a), which is likely mediated by Ant1, since it was sensitive to carboxyatractyloside.

Reviewer #2,

A. The question raised here is to explain the findings that the ANT deletion in liver has not the expected life threatening effects given the central role of the ANT in ATP supply. Mitochondria remain intact h on deletion ANT. Interestingly, the synthesis of mitochondria is even increased. Since respiration with substrates such as pyruvate is normal, an uncoupling has to be invoked. Since the membrane potential is not decreased the uncoupling has to be subtle. Uncoupling proteins are not involved. The nature of the uncoupling remains unknown.

A major disease related aspect of this paper are the fat metabolism changes associated with the ANT levels. Fatty liver disease on over-nutrition is suppressed by ANT deletion. A similar prevention of the fatty liver or even of general obesity has the injection of the inhibitor CAT in normal mice.

B. This is an interesting paper where paradoxical phenomena challenge conventional wisdom.

C. ok

D. ok

E. The conclusions should be modified according to forthcoming results (see F)

F. inhibitor CAT in normal mice.

The following amendments are suggested. In order to understand the status of the ANT deleted mitochondria the content of ATP synthase (AS) is needed. AS activity should decrease in ANT depleted mitochondria. On the other hand AS might be responsible for the uncoupling. For example if ANT forms super-complexes with AS. The lack of ANT might destabilize the AS dimers which are known to form pores in the "permeability transition". Thus AS might play a key role for understanding the ANT deficient mitochondrial metabolism

G. ok

H. ok

We newly tested the effects of oligomycin on respiration of Ant2-depleted liver mitochondria using Seahorse XF24 (**Fig. 3d**). Interestingly, oligomycin did not affect their basal respiration status; thus the data implied that the F1F0 ATPase is unlikely involved in proton leak here. These data are now included and discussed in the revised manuscript.

Reviewer #3,

Cho et al analyze hepatocyte-specific ANT2 null mice, which exhibit low body weight, protection from HFD induced adiposity, and protection from fatty liver. These phenotypes are associated with hepatocyte mitochondrial proliferation, abrogated ADP-stimulated respiration, and indirect evidence of uncoupled respiration that does not require UCP2. The authors suggest ANT2 may be a target for NAFLD. The studies are interesting, but would improve from modest experimental revision. More confirmation of phosphorylation-independent proton leak, driving increased substrate oxidation, is needed.

1. *Uncoupled respiration has been supported, but not demonstrated, because mitochondrial number is increased, and state 2 respiration is normalized to mitochondrial mass. A respiration experiment in liver tissue or isolated hepatocytes should be performed to demonstrate increased state 2 respiration.*

We agree that it would be nice to demonstrate an increase in overall respiration *per cell*. However, unfortunately we were not able to measure respiration of perfused hepatocytes due to technical difficulties. However, to confirm the data shown **Fig. 3c**, we newly measured oxygen consumption rate of liver mitochondria using a Seahorse XF24 as well. As shown **Fig. 3d**, we demonstrated that basal respiration per mitochondria mass is similar or slightly higher in cKO, with no response to ADP at all. We hope these along with the following *phosphorylation-independent* respiration data would strengthen our discussion here.

2. *Formal measurements of proton leak should be made, e.g. as performed in Cell Metab. 2006;3:417-27. Membrane potential appears to be normal in ANT2KO. To confirm proclivity toward uncoupling, rates of membrane potential-dependent proton leak should be measured using escalating malonate concentrations (which collapse membrane potential).*

We did measure *phosphorylation-independent* respiration and responses to malonate using an O2k respirometer. The Ant2-deleted liver mitochondria showed a significantly higher rate of *phosphorylation-independent* respiration and lower sensitivity to malonate. These new data are now included as **Fig. 3e** and **Extended Data Fig. 3c**. It should be noted, however, that we were not able to measure membrane potential simultaneously since we do not have a multi-sensor for the purpose.

3. *The hypothesis that F1F0 ATPase may favor proton leak and ATP hydrolysis is attractive, and easy to assess using oligomycin. If it is not the F1F0 ATPase, then there are other potential candidates.*

We newly tested the effects of oligomycin on respiration of Ant2-depleted liver mitochondria using Seahorse XF24 (**Fig. 3d**). Interestingly, oligomycin did not affect their basal respiration status; thus the data implied that the F1F0 ATPase is unlikely involved in proton leak here. These data are now included and discussed in the revised manuscript.

4. *Liver is typically 20% of resting energy expenditure. It will almost certainly be much higher in these animals. At the very least, whole animal VO₂/energy expenditure must be measured.*

We tested whole-body energy expenditure of the mice under a high fat diet using a CLAMS. Although it tended to be slightly higher in the *Ant2* cKO mice, there was no statistical difference. These new data are now included in the manuscript as **Fig. 5d**, and discussed in the text.

5. *The source of the Ant2fl allele is not clear. Is this new or the same as that in Nature 427, 461-465 (2004)? If it's the same allele, then Fig. 1a should move to supplemental. If it's a new construction, then more details should be given in the methods.*

The mice have been originally generated by us and are different from those Doug Wallace's group has described (*Nature* **427**, 461, 2004). Exons 2 & 3 are floxed in our mice, whereas Exons 3 & 4 are floxed in theirs. Since we have described our *Ant2* gene targeting strategy more in details at our previous publication on systemic Ant2 hypomorphic mice (*Cell Death Differ* **22**, 1437, 2015), we provided minimal information here. In addition, we are currently depositing our *Ant2* cKO mice to the Jackson Lab to make them available to other investigators. This information is now included at the method section.

6. *Give body weights on HFD.*

We now provide such data at **Fig. 5a**. Furthermore, we performed more comprehensive analyses regarding the effects of Ant2 cKO on HFD-induced obesity and related metabolic changes. We included these results as an independent figure (**Fig. 5**).

7. *The term pseudo-fasting should be eliminated. This is not at all the fasting state, even with weight loss and mild ketosis.*

We removed the term from the text.

We thank all the reviewers again for their very helpful and constructive criticisms. We do believe that the revised manuscript has been significantly improved by incorporating their suggestions and advice.

Reviewers' comments:

Reviewer #1 (Remarks to the Author):

My concerns were addressed

Reviewer #2 (Remarks to the Author):

The author has responded to a large part of the critical comments. However, the suggestion to examine the role of ATP synthase (AS) in the uncoupling, which is a central issue raised by reviewer 2, has not been properly answered. It is well established that AS is involved in the "permeability transition" (PTP) of mitochondria, promoted by cyclophilin and inhibited by cyclosporin A. The pore is formed by destabilization of the normal functioning AS dimers. The lack of ANT could be the cause for the destabilization. But different from the author statement on line 168 this type of uncoupling is not associated or caused by ATP hydrolysis. In the new experiments the authors show that uncoupling is not inhibited by oligomycin, concluding that AS is not involved. However the reported lack of respiratory response to oligomycin (fig 3 d) does not argue against an AS involvement in the observed uncoupling - at variance to the author's conclusion. Instead, it would have been important to test the influence of cyclosporin A on the uncoupling to discern the PTP involvement.

Reviewer #3 (Remarks to the Author):

The authors have adequately addressed my concerns. The experiment presented in Fig. 3e, revealing increased proton leak in Ant2 KO mitochondria, was compelling. The authors describe this as 'A reduced sensitivity to malonate...' but actually reveals intact responsiveness to malonate (slope is the same), with a reset baseline in proton leak.

Two features remain unclear -

1. The mechanism for uncoupling, which now appears to be F1F0-ATPase independent. This mechanism may relate to the mitochondrial proliferation that is evident - a topic for future study.
2. The leanness of Ant2 cKO mice. This is probably attributable to a subtle increase in energy expenditure that the calorimetry method used is missing. The authors might consider dividing their analysis between dark and light cycles to reveal a greater difference.

RESPONSE TO REVIEWERS

Please find point-by-point responses to the reviewer's comments below.

Reviewer #2,

The author has responded to a large part of the critical comments. However, the suggestion to examine the role of ATP synthase (AS) in the uncoupling, which is a central issue raised by reviewer 2, has not been properly answered. It is well established that AS is involved in the "permeability transition" (PTP) of mitochondria, promoted by cyclophilin and inhibited by cyclosporin A. The pore is formed by destabilization of the normal functioning AS dimers. The lack of ANT could be the cause for the destabilization. But different from the author statement on line 168 this type of uncoupling is not associated or caused by ATP hydrolysis. In the new experiments the authors show that uncoupling is not inhibited by oligomycin, concluding that AS is not involved. However the reported lack of respiratory response to oligomycin (fig 3 d) does not argue against an AS involvement in the observed uncoupling - at variance to the author's conclusion. Instead, it would have been important to test the influence of cyclosporin A on the uncoupling to discern the PTP involvement.

Thank you for your very useful suggestions. Since the KO liver did not show any sign of cell death, we considered that PTP is unlikely involved here. However, in order to exclude the possibility, we newly measured mitochondrial respiration before and after addition of Cyclosporin A (CsA) using a Seahorse analyzer. As shown in Supplementary figure 3e, respiration was not affected by addition of CsA both in wild type and cKO liver mitochondria. These newly added data indicate that PTP, which could be triggered by AS changes, is not involved in the observed uncoupling.

REVIEWERS' COMMENTS:

Reviewer #2 (Remarks to the Author):

The authors have performed the controls to exclude the mitochondrial transition pore as a mechanism of uncoupling and thus satisfied the reviewer`s 2 requests.
Recommended acceptance.